# Preparation and Investigation of a Nanosized Piroxicam Containing Orodispersible Lyophilizate

**DOI:** 10.3390/mi15040532

**Published:** 2024-04-15

**Authors:** Petra Party, Sándor Soma Sümegi, Rita Ambrus

**Affiliations:** Institute of Pharmaceutical Technology and Regulatory Affairs, Faculty of Pharmacy, University of Szeged, Eötvös u. 6, 6720 Szeged, Hungary; party.petra@szte.hu (P.P.); sumegi.s.soma@gmail.com (S.S.S.)

**Keywords:** piroxicam, nanoprecipitation, solvent diffusion, freeze drying, orodispersible, lyophilizate, analgesics

## Abstract

Non-steroidal anti-inflammatory piroxicam (PRX) is a poorly water-soluble drug that provides relief in different arthritides. Reducing the particle size of PRX increases its bioavailability. For pediatric, geriatric, and dysphagic patients, oral dispersible systems ease administration. Moreover, fast disintegration followed by drug release and absorption through the oral mucosa can induce rapid systemic effects. We aimed to produce an orodispersible lyophilizate (OL) consisting of nanosized PRX. PRX was solved in ethyl acetate and then sonicated into a poloxamer-188 solution to perform spray-ultrasound-assisted solvent diffusion-based nanoprecipitation. The solid form was formulated via freeze drying in blister sockets. Mannitol and sodium alginate were applied as excipients. Dynamic light scattering (DLS) and nanoparticle tracking analysis (NTA) were used to determine the particle size. The morphology was characterized by scanning electron microscopy (SEM). To establish the crystallinity, X-ray powder diffraction (XRPD) and differential scanning calorimetry (DSC) were used. A disintegration and in vitro dissolution test were performed. DLS and NTA presented a nanosized PRX diameter. The SEM pictures showed a porous structure. PRX became amorphous according to the XRPD and DSC curves. The disintegration time was less than 1 min and the dissolution profile improved. The final product was an innovative anti-inflammatory drug delivery system.

## 1. Introduction

The quantity of newly researched medications, the sensitivity to clinical findings, and the increasing cost of healthcare have all contributed to the growing need for innovative drug delivery approaches. The majority of available and newly created medication compounds are poorly soluble, which can cause serious challenges with formulation and low bioavailability. New strategies can be used to solve this problem, pointing toward nanotechnology and alternative drug delivery systems [1].

The non-steroidal anti-inflammatory drug (NSAID) piroxicam (PRX), a class II medication according to the Biopharmaceutical Classification System (BCS), was selected for the work. Its oral absorption is considered to be limited by its dissolution rate. PRX has a well-established safety profile because it has been used in marketed products for several decades in a variety of dose forms and administration methods, including oral [2]. When administering PRX compared to other NSAIDs, only low doses (20 mg/day) are necessary to achieve a therapeutic effect. PRX also shows deep localized tissue penetration, which is mainly delivered through systemic circulation. However, it is ulcerogenic due to its slow and gradual absorption [3]. The development of new formulations can improve its absorption and provide rapid relief in acute and chronic musculoskeletal and joint problems such as ankylosing spondylitis, rheumatoid arthritis, and osteoarthritis [4].

Nanosizing is one of the best methods to improve the solubility and dissolution rates of poorly water-soluble active pharmaceutical ingredients (APIs). One of the most promising dosage forms in recent years has been the dispersions of nanoparticles stabilized with the help of polymers or surfactants, known as nanosuspensions (NSs) [5]. Reducing PRX particles to nano size by increasing the specific surface area is a solution to accelerate the dissolution rate, thereby increasing bioavailability. Numerous techniques have been explored for creating NSs of drugs that are poorly water-soluble [6,7]. These methods can be divided into two main categories: bottom-up methods start from the molecular level to build the crystal, while top-down methods involve the crushing of large particles. In this study, we used the solvent diffusion method to create nanoparticles. The method consists of dissolving the drug in its solvent and then mixing it with an anti-solvent. As a consequence, the active substance precipitates from its dissolved state. An important prerequisite for this method is that the two solvents are slightly soluble [8,9,10].

The solidification of the NS could increase its stability and enable the application of various delivery routes. Recently, there has been an increasing interest in the development of orodispersible dosage forms (ODFs), and many drug products have already been approved, such as oral lyophilizates, orodispersible tablets, orodispersible granules, and orodispersible films [11]. The pharmaceutical industry invests in ODFs because they are a simple way to broaden their product line and can be marketed as line extensions if their bioequivalence is established [12]. The primary benefit of these dosage forms is their rapid disintegration in the oral cavity from the solid state to a solution or suspension; as a result, they can improve patient compliance when administering drugs orally to particular patient populations (pediatric, geriatric, dysphagic, and psychiatric patients) [13,14]. Water is not required for administration; the volume of saliva is sufficient for complete disintegration [15]. The active component can be released locally in the oral cavity and absorbed, either directly through the oral mucosa, where the rapid onset of action is expected, or after swallowing through the intestinal barrier [16,17].

Oral lyophilizates (OLs) are considered a type of ODT due to their similar shape and packaging [18]. An OL is defined by the European Pharmacopoeia as a solid dosage form that is freeze-dried, designed to rapidly disintegrate in the mouth [19]. Usually, the freeze-drying process takes place directly in the blister, which is also the primary packaging for the final dosage form [11]. The freeze-drying method is frequently employed to extend the shelf life of certain products or to dry thermolabile materials. It can also be employed in the preparation of oral dosage forms in the shape of tablet-like porous matrixes, with a very short disintegration time [20]. The most frequently used additives in the formulation of OLs are matrix-forming excipients, such as gelatin, alginates, and amino acids; these maintain the structural integrity of the matrix even after the liquid components are removed. Surfactants guarantee proper wetting upon administration, such as poloxamer, while thickening agents, for example, hydroxypropyl methylcellulose or xanthan gum, improve the uniformity of the formulation. Bulking agents may also bring additional advantages of cryoprotection and taste masking, as is the case of mannitol, one of the most widely used excipients for this application. Nevertheless, disintegrants are excipients that promote disintegration by promoting liquid penetration into the OL and then expanding upon hydration, for example, croscarmellose sodium or alginates [18,21].

We aimed to produce an OL consisting of nanosized PRX to synthesize the advantages of the nanoparticles and ODF. A combined preparation method was used; after spray-ultrasound-assisted solvent diffusion-based nanoprecipitation, the solid form was freeze-dried using appropriate additives [10,16]. We were expecting fast disintegration and drug dissolution from the OL containing PRX. The final goal was an orodispersible formulation that could be useful in the pain therapy of patients with different forms of arthritis.

## 2. Materials and Methods

### 2.1. Materials

Piroxicam (PRX, Secifarma, Milano, Italy) was the active ingredient. Ethanol (EtOH, Molar Chemicals Kft., Halásztelek, Hungary) and ethyl acetate (EtAc, Molar Chemicals Kft., Halásztelek, Hungary) were used as solvents. Poloxamer-188 (POL, Sigma-Aldrich Co., Ltd., Budapest, Hungary), D-Mannitol (MAN, Sigma-Aldrich Co., Ltd., Budapest, Hungary), and sodium alginate (NaAlg, Sigma-Aldrich Co., Ltd., Budapest, Hungary) were applied as excipients during freeze-drying.

### 2.2. Preparation Method

#### 2.2.1. Selection of the Organic Solvent

PRX is a very poorly soluble compound in water (0.0023 mg/mL at 22 °C [22]); therefore, it was chosen as an anti-solvent during the solvent diffusion based nanoprecipitation. The selection of a good solvent was based on the data from the literature and experimental determination. During selection, we took into consideration the fact that the solvent should fulfil the requirements for modern green solvents. Therefore, the solubility tests of PRX in EtOH and EtAc at room temperature were carried out using the gravimetric method. PRX was added to the solvents at room temperature until crystals appeared. To ensure that the solutions were saturated, they were stirred with a magnetic stirrer (AREC.X heating magnetic stirrer, Velp Scientifica Srl., Usmate Velate, Italy) for another 24 h. From the supersaturated crystalline suspension, 1 mL of samples were taken with a pipette and measured in preweighed Petri dishes. The Petri dishes were placed in a drying chamber. After complete drying, the weight of the solved PRX in 1 mL of organic solvent was determined. For both solvents, three parallel measurements were performed. PRX was slightly soluble in EtOH 0.61 mg/mL; therefore, EtAc was chosen as a solvent (12.04 mg/mL).

#### 2.2.2. Optimization of the Process Parameters

To select the concentration of the stabilizer in the anti-solvent, the concentration of the PRX in EtAc, and the settings of the spray ultrasound sonotrode, preliminary experiments were performed. The Plackett and Burman screening design was chosen, which yields unbiased estimates of all main effects in the smallest design possible. Thus, PBD requires fewer experiments than the highly fractionated factorial designs that include the same number of factors [23]. Seven factors were investigated at two different levels using TIBCO Statistica^®^ 13.4 software (Statsoft Hungary, Budapest, Hungary). The PRX (5 and 10 mg/mL), concentration of stabilizer polymer concentration (0.1 and 0.2 m/m%), time of the sonication time (2 and 4 min), amplitude (50 and 100%), cycle (50 and 100%), power (50 and 100 W), and velocity of the pump (20 and 40 mL/min) were the relevant factors. The measured responses were as follows: average hydrodynamic diameter (Z average), polydispersity index (PI), and zeta potential (ζ). The most significant factors were determined.

#### 2.2.3. Spray-Ultrasound-Assisted Solvent Diffusion-Based Nanoprecipitation

According to the results of the factorial design, the final parameters were as follows: 50 mg of PRX was dissolved in 9.95 g of EtAc, then sonicated in 90 g of 0.2% (*w*/*w*) POL solution with purified water (UP200St Powerful Ultrasonic Lab Homogenizer, Hielscher Ultrasonics GmbH, Teltow, Germany) for 4 min. The following settings were applied: pump: 40 mL/min, power: 100 W, cycle: 100%, amplitude: 100%. The result of the process was a NS (Figure 1).

#### 2.2.4. Lyophilization

The NS consisting of PRX was freeze-dried using a freeze dryer (Lyovapor L-200 Pro, Büchi, Flawil, Switzerland). The final composition was 40 g of PRX NS, 2 g of MAN, and 0.5 g of NaAlg (Table 1). Then, 1 mL of the sample was taken from the NS and poured into blister sockets. The blisters were placed on the freeze dryer shelf and cooled to −54 °C. The pressure was reduced to 1 mbar and was kept constant to 6 h for the complete solidification of the product (Figure 1).

### 2.3. Particle Size Analysis

#### 2.3.1. Dynamic Light Scattering (DLS)

A Malvern Zetasizer Nano ZS (Malvern Instruments, Worcestershire, UK) was used to determine the average hydrodynamic diameter (Z average), polydispersity index (PI), and zeta potential (ζ pot.) via DLS. The NS was loaded into folded capillary cells and measured at 25 °C. A refractive index of the PRX was set to 1.632. Three investigations were carried out.

#### 2.3.2. Nanoparticle Tracking Analysis (NTA)

The NanoSight NS 3000 device (Malvern Instruments, Worcestershire, UK) for NTA was used to obtain high-resolution particle size information. The instrument was equipped with a 565 nm laser, a high sensitivity sCMOS camera, and a syringe pump. The MX suspension was diluted 1000 times and loaded into the device using a syringe pump speed of 50. The experiment videos were analyzed using NTA 3.4 Build 3.4.4 after capture in script control mode (3 videos of 30 s per measurement). A total of 1500 frames were examined. The particle size distribution was characterized by values of D10 (50% of particles were below this diameter), D50 (50% of the particles were below this diameter), and D90 (90% of the particles were below this diameter). The particle size distribution (PSD) was calculated according to the following equation:(1)$$Span=\frac{D90-D10}{D50}$$

### 2.4. Surface Tension Investigation

The liquid–gas interfacial tensions for three fluids (POL solution, POL-EtAc solvent mixture, and NS) were characterized using the pendant drop technique of the OCA 20 apparatus (Dataphysics Instrument GmbH, Filderstadt, Germany). The balance between the forces of gravity and interfacial adhesion, which binds the droplet to the needle, provides the foundation for these observations. An injection needle was used to generate a drop of fluid. The integrated camera took pictures of the drop at 25 °C. SCA 20 software was used to record the geometry of the drop and the surface tension of the fluids was determined using the Young–Laplace equation. The density values of the samples were measured prior to the investigation. Ten pictures were obtained for each experiment, and the average surface tension was calculated.

### 2.5. Morphology Investigation

Employing scanning electron microscopy, the size and form of the particles were evaluated (SEM, Hitachi S4700; Hitachi Ltd., Tokyo, Japan), operating at a voltage of 10 kV. Using a sputter coater (Bio-Rad SC502; VG Microtech, Uckfield, UK) and a 2.0 kV electric potential at 10 mA for 10 min, the samples received a gold–palladium coating. The air pressure was between 1.3 and 13.0 mPa. ImageJ, a freely accessible image analyzer (https://imagej.nih.gov/ij/index.html, accessed on 10 February 2024), was used to calculate the diameter of the PRX particles in case of the dried NS and the lyophilizate.

### 2.6. Analysis of the Structure

#### 2.6.1. Differential Scanning Calorimetry (DSC)

The structures of the products were examined using a Mettler Toledo TGA/DSC thermal analysis device (Mettler-Toledo GmbH, Greifensee, Switzerland). The DSC measurements were performed by investigating 3 to 5 mg of samples heated to temperatures of 25 to 300 °C at a rate of 10 °C/min while maintaining a flow of argon at a rate of 10 L/h. The STARe program was used to evaluate the data (Mettler-Toledo GmbH, Greifensee, Switzerland).

#### 2.6.2. X-ray Powder Diffraction (XRPD)

The BRUKER D8 advance X-ray powder diffractometer (Bruker AXS GmbH, Karlsruhe, Germany) and the VNTEC-1 detector (Bruker AXS GmbH, Karlsruhe, Germany) were applied to carry out the XRPD measurements. The powder samples were placed on a slide of flat quartz glass with an etched square. The samples were measured at 40 kV and 40 mA with a step period of 0.1 s and a step size of 0.007°, and the angular range was 3–40°. The DIFFRACplus EVA program was used for all adjustments, including K2 stripping, background removal, and smoothing of the area under the peaks.

### 2.7. Fourier-Transform Infrared Spectroscopy (FT-IR)

The interactions between PRX and the excipients were investigated using an AVATAR 330 FT-IR spectrometer (Thermo Nicolet, Thermo Fisher Scientific Inc., Waltham, MA, USA). The raw materials were combined into physical mixtures (PM) with the same composition as OL. The samples were homogenized with 150 mg of KBr in an achate mortar, and the mixture was pressed to make pastilles using a Specac^®^ hydraulic press (Specac, Inc., Washington, PA, USA) by 10 ton pressing force. The infrared spectra were recorded between 4000 and 400 cm^−1^, at an optical resolution of 4 cm^−1^.

### 2.8. In Vitro Disintegration Test

To demonstrate the orodispersability of lyophilizate, an in vitro disintegration test was performed in 200 mL of purified water at 20 ± 0.5 °C. Using a digital stopwatch, the amount of time required for total disintegration was observed, that is, until no solid residue was detected. The average disintegration time and standard deviation of six tested OLs were determined.

### 2.9. In Vitro Drug Release Study

Using a modified paddle method from the European Pharmacopeia [24], the rate of drug release of the powders was examined (Hanson SR8 Plus, Teledyne Hanson Research, Chatsworth, CA, USA). The Pharmacopoeia does not contain any specification for orally dispersible systems, so the release medium to mimic saliva was prepared based on the literature [25]. The analysis was carried out under buccal circumstances (37 °C and pH 6.8). The following components were included in 50 mL of artificial saliva: 8.00 g/L NaCl, 2.38 g/L NaH_2_PO_4_, and 0.19 g/L KH_2_PO_4_. The paddle rotated at 50 rotations per minute. The sample preparation was completed after 5, 10, 15, and 30 min. To maintain the constant permanent volume, saliva was simultaneously added at each sampling point to replace 2 mL of the sample. For filtering, cellulose ester membranes with 0.45 μm pore sizes were employed. Following filtration, spectrophotometry at 355 nm (Unicam UV/VIS Spectrophotometer, Cambridge, UK) was used to determine the drug content of the aliquots. The experiments were carried out in three sets. The investigation was performed in the case of the raw PRX and the OL containing nanosized PRX, to compare their drug release.

## 3. Results

### 3.1. Outcomes of the Particle Size Analysis

#### 3.1.1. Dynamic Light Scattering Results

The original particle diameter of the dry PRX particles was around 9.38 µm according to previously performed laser diffraction measurements. The DLS investigations showed that the Z average of the drug decreased into the nano range. The particle diameter was under 200 nm, which predicted the improved dissolution properties of the PRX under buccal conditions. The PI was 0.33, which indicated a slight polydispersity (>0.3) in the NS. Therefore, the proportion of small particles may be underestimated due to the fact that larger particles scatter more light than smaller particles during the DLS test [26]. Consequently, further particle size characterization was performed (Section 3.1.2) to more precisely investigate particle diameter and size distribution. The ζ-potential results showed that the particles in the suspension were at the beginning of agglomeration [27], but this was not considered a problem because the NS remained stable until the following freeze-drying process (Table 2).

#### 3.1.2. Nanoparticle Tracking Analysis Results

The size analysis of the NS was also performed using NTA, which is comparable to DLS and successfully minimizes its drawbacks. This technique used nanoparticle light scattering and Brownian motion to determine the size distribution of the sample. NTA simultaneously detects large and small particles and considers them individual particles, resulting in a more accurate particle distribution [26,28,29]. The diameter of the particles measured here was approximately 120 nm (Figure 2, Table 3), which is suitable for enhanced drug delivery through the oral mucosa. Nevertheless, the evaluated Span value was 0.83. Therefore, the particle size distribution was monodispersed (Span < 2), which is highly advantageous in case of NS, and also in the solid form to implement the proper dosing.

### 3.2. Surface Tension Investigation

The surface tension (ST) of the 0.2% (*w*/*w*) POL solution was 53.63 ± 0.18 mN/m. Due to its surfactant properties, the polymer decreased the ST of the water (71.99 ± 0.36 mN/m at 25 °C) [30]. The ST of EtAc is 23.97 ± 0.10 mN/m at 25 °C [31]. The ST of the combination of POL solution and EtAc was 49.36 ± 0.45 mN/m, which predicted that the solvent mixture is capable of stabilizing NS [32]. The presence of PRX nanoparticles increased this ST value to 55.35 ± 4.21 mN/m.

### 3.3. Morphology Investigation

To better understand the morphological properties, SEM images of the oven-dried NS and the OL were taken. The SEM pictures showed nanosized PRX particles in the case of the dried NS and the OL (Figure 3). According to the ImageJ analyses, the PRX particles were larger compared to the DLS and NTA measurements due to the POL forming a matrix around the drug; moreover, the possibility of a small degree of aggregation during drying can not be excluded (Table 4). However, this particle size, around 350 nm, was significantly reduced compared to the raw API. Additionally, the POL coating of the PRX particles helps to maintain the individuality of the nanoparticles and can promote faster dissolution of the drug in saliva. On the image of the OL, a porous structure was observable, which was the result of the application of MAN and NaAlg. They formed a proper matrix and added bulk to the lyophilizate to maintain a tablet-like solid dosage form, which will quickly disintegrate in the mouth.

### 3.4. Results of the Crystallinity Investigation

#### 3.4.1. Differential Scanning Calorimetry Result

DSC measurements were performed to study changes in the crystalline structure of the materials. During nanoprecipitation, the PRX became amorphous and remained this way during freeze-drying according to the DSC curves (Figure 4). This showed that the nanoprecipitation caused the amorphization of the PRX and that it did not recover its crystalline structure until the end of the lyophilization. The POL retained its crystallinity after the nanoprecipitation; however, it became amorphous after freeze-drying. In addition, it can be seen that the crystalline structure of the applied additives changed after freeze-drying. The melting point of NaAlg was not observable and the structure of the MAN changed. Before processing, the beta modification of the MAN was transformed into a delta modification, which was reflected in the change in the melting point [33].

#### 3.4.2. X-ray Powder Diffraction

The characteristic peaks of the originally crystalline raw materials, PRX, POL, and MAN, were observable (Figure 5). The original solid PRX was in its anhydrate form according to the literature [34]. The amorphization of the PRX reported from the DSC curves was confirmed by XRPD analysis. The changes in the crystalline structure of MAN can be seen when comparing the peaks of initial MAN with the freeze-dried sample.

### 3.5. Fourier-Transform Infrared Spectroscopy

FT-IR analysis was carried out to study the possible molecular interactions between the PRX and the excipients (POL, MAN, NaAlg). To identify the changes occurring during nanonization and freeze-drying, the FT-IR spectra of the original PRX and MAN, the PM, and the OL were compared (Figure 6). However, the ratios of the materials, in the case of PM and OL, resulted in the PRX being not well observable due to the presence of MAN dominating the spectra. The shifts of the spectra of MAN after lyophilization could be due to the change in its structure [33], as reported by DSC and XRPD.

### 3.6. In Vitro Disintegration Test

To prove the mouth dispersibility of the final product, a disintegration study test was carried out. The maximum disintegration time specified in the European Pharmacopoeia is 3 min [19,35]. The produced OL took 44.5 ± 8.83 s to disintegrate, which was the result of the appropriate ratio of MAN and NaAlg. Furthermore, the incorporation of MAN can improve the taste of the product. On the basis of this in vitro study, it can be concluded that the lyophilizate can be utilized as a suitable orodispersible system.

### 3.7. In Vitro Drug Release Study

To demonstrate the efficacy of the nanosized API, in vitro drug dissolution was also tested. The requirements for the immediate release dosage forms are that at least 80% of the API should be released in the specified time, usually 45 min or less [36]. Thanks to the improved surface area, the dissolution of the nanosized PRX was rapid compared to the initial form of the drug (Figure 7). From the dissolution curve, it can be seen that 80% of the nanosized API was dissolved from the OL after 5 min thanks to the reduced particle size of PRX, the applied hydrophilic excipient, and the added disintegrant. This ensures rapid absorption and the rapid onset of action. In the literature, there are examples of the increased dissolution rate leading to more rapid absorption and improved drug delivery performance compared to standard-reference PRX tablets supported by a clinical study [2].

## 4. Discussion

This article describes the successful formulation of an orodispersible lyophilizate containing nanosized PRX. A simple and cost-effective strategy for formulating NS involved using a high-performance spray-ultrasound sonication-assisted solvent diffusion method to prepare NSs of the poorly water-soluble PRX. The efficacy of the formulation was determined by characterizing the physical, chemical, and structural properties of the NS and performing dosage form studies. The particle diameter was lower than 200 nm according to DLS and NTA investigations. The NS was further freeze-dried with the addition of excipients (MAN, NaAlg). The method resulted in a nanosized PRX containing OL. The SEM images showed the nanosized PRX and the advantageous porous structure of the OL. Two different analytical methods (XRPD and DSC) indicated the amorphization of the PRX. OL disintegrated in 1 min and PRX was released rapidly during the in vitro dissolution test due to the increased surface area and the amorphous form of API. The result was an innovative anti-inflammatory drug delivery system for systemic effect. Additionally, based on the results obtained, supergenerics could be developed in the future using this technology.

## Figures and Tables

**Figure 1 micromachines-15-00532-f001:**
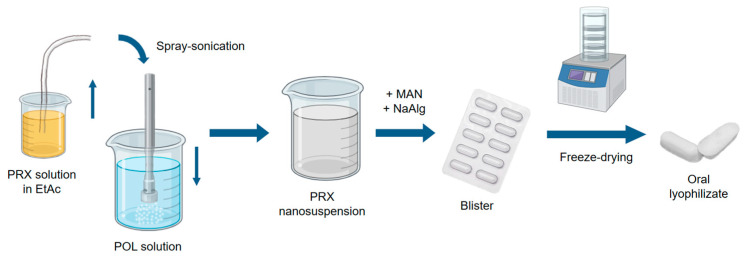
Schematic figure of the preparation method (created using BioRender.com, accessed on 12 February 2024.).

**Figure 2 micromachines-15-00532-f002:**
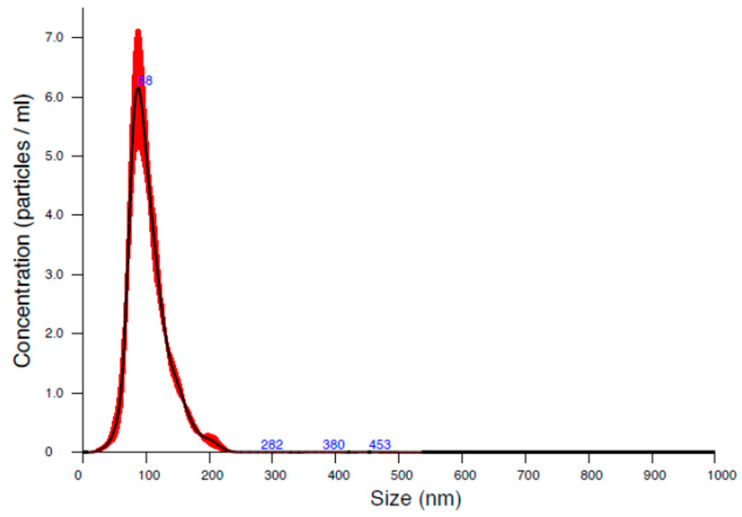
NTA results of the NS (red color shows the distribution of the particles, blue color shows the size of the particles at exact points.

**Figure 3 micromachines-15-00532-f003:**
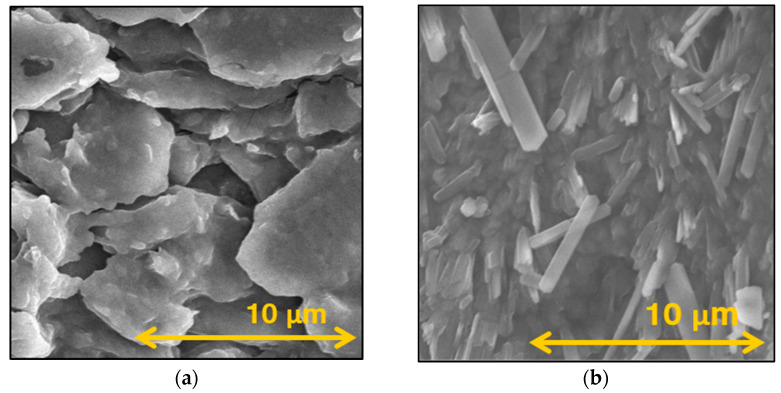
SEM images of the NS (**a**) and the OL (**b**).

**Figure 4 micromachines-15-00532-f004:**
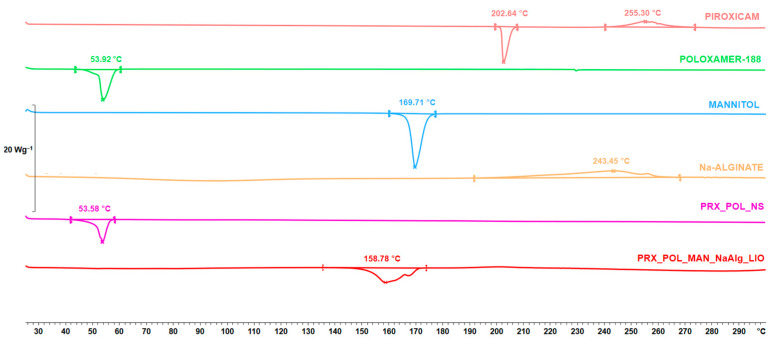
DSC results of the initial materials, the NS, and the OL.

**Figure 5 micromachines-15-00532-f005:**
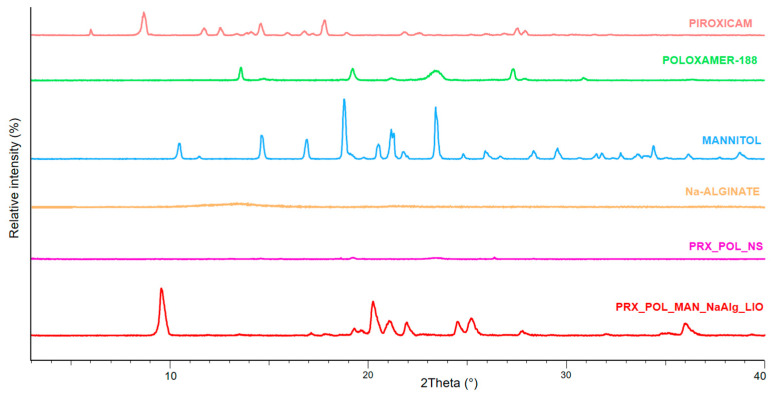
XRPD results of the initial materials, the NS, and the OL.

**Figure 6 micromachines-15-00532-f006:**
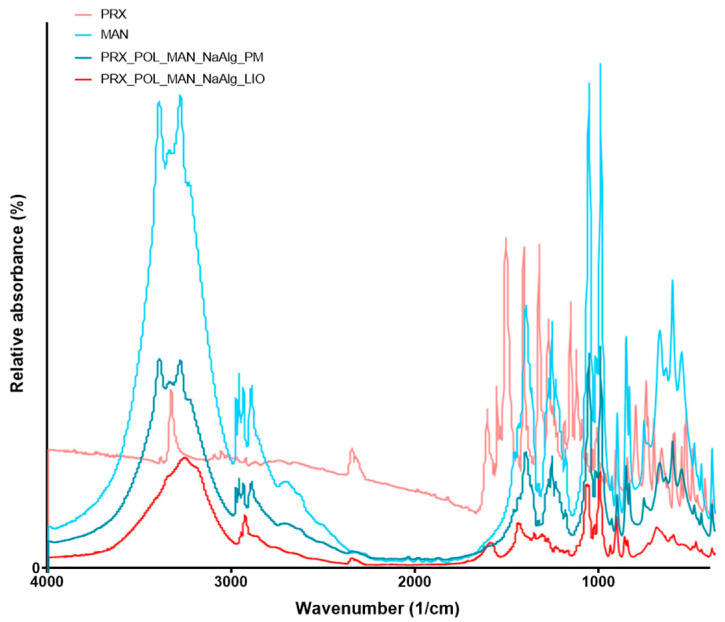
FTIR results of two initial materials: the PM and the OL.

**Figure 7 micromachines-15-00532-f007:**
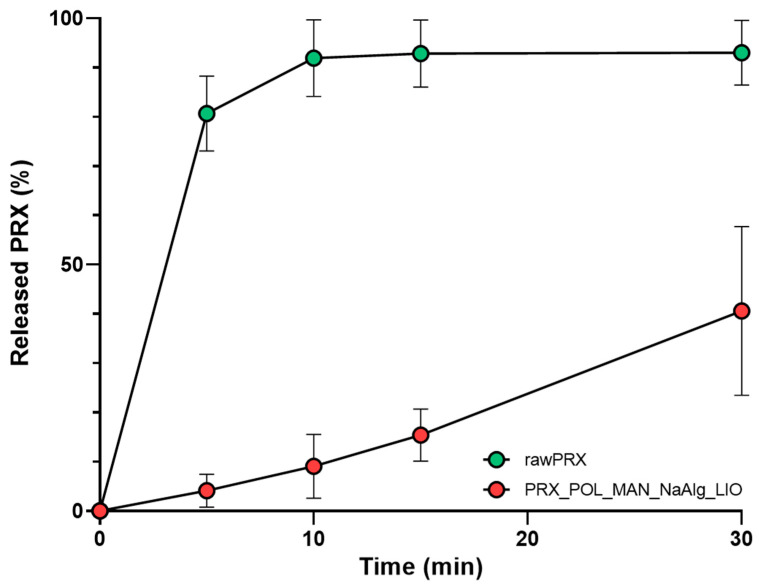
Dissolution of the initial drug and the OL. Data are means ± SD (*n* = 3 independent measurements).

**Table 1 micromachines-15-00532-t001:** Composition of the NS and the OL.

Sample	PRX (mg)	POL (mg)	MAN (mg)	NaAlg (mg)
PRX_POL_NS	50	180	-	-
PRX_POL_MAN_NaAlg_LIO	0.5	2	50	12.5

**Table 2 micromachines-15-00532-t002:** DLS results of the NS.

Sample	Z Average (nm)	PI	ζ Potential (mV)
PRX_POL_NS	185.03 ± 0.76	0.33 ± 0.05	−12.37 ± 0.46

Data are means ± SD (*n* = 3 independent measurements).

**Table 3 micromachines-15-00532-t003:** NTA results of the NS.

Sample	d (nm)	D10 (nm)	D50 (nm)	D90 (nm)
PRX_POL_NS	119.4 ± 0.8	90.6 ± 1.2	110.6 ± 0.6	159.9 ± 6.1

Data are means ± SD (*n* = 3 independent measurements).

**Table 4 micromachines-15-00532-t004:** ImageJ particle size analysis according to SEM pictures.

Sample	d (nm)
PRX_POL_NS	373.73 ± 86.48
PRX_POL_MAN_NaAlg_LIO	348.17 ± 73.54

Data are means ± SD (*n* = 100 independent measurements).

## Data Availability

The original contributions presented in the study are included in the article. Further inquiries can be directed to the corresponding author.

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
