# Peer review of "Preparation and Investigation of a Nanosized Piroxicam Containing Orodispersible Lyophilizate"

_micromachines, 2024, doi:10.3390/mi15040532_

Round 1

Reviewer 1 Report

Comments and Suggestions for Authors

The manuscript titled "Preparation and investigation of a nanosized piroxicam-containing orodispersible lyophilizate" by Rita et al. attempts to address the poor solubility of Piroxicam (PRX) by formulating a nanosized PRX-containing orodispersible lyophilizate.

- In section, line 101, material description for ethanol is not mentioned.

- In Line 109, the authors mentioned the solubility in water in mg/L. Please keep the same units consistent throughout the document either mg/mL or mg/L.

- In line 109, add a reference for the PRX water solubility.

- In line 129, remove the open bracket and correct the sentence to include all the relevant factors that were investigated.

- In line 137, does it mean 50mg of PRX 'dissolved' (not solved) in 9.95g of ethyl acetate? Please clarify/correct.

- In line 164, was the pump speed set at 50ml/min? Please clarify.

- In line 305, Figure 5, upon closer observation, the red-colored PRX_POL_LIO (bottom), Piroxicam (Top), and POL (2nd), the last one seems not a physical mixture of PRX+POL; it appears to be a polymorph for me. My question that needs clarification here is: Did the solubility increase due to this new form, or due to nanosizing with Lyophilization? Check the literature for any existing PRX  polymophs matching with the current new pattern in the figure to make sure it is not a new polymorph. If it is any existing polymorph then you have to explain that properly in the manuscript.

Author Response

Thank you very much for your opinion. Below are listed all of the modifications made in the paper, according to the suggestions, and you can find them with purple color in the text.

In section, line 101, material description for ethanol is not mentioned.

Thank you for your comment. The text was modified.

In Line 109, the authors mentioned the solubility in water in mg/L. Please keep the same units consistent throughout the document either mg/mL or mg/L.

Thank you for your comment. The text was modified.

In line 109, add a reference for the PRX water solubility.

Thank you for your comment. The reference was added.

In line 129, remove the open bracket and correct the sentence to include all the relevant factors that were investigated.

Thank you for your comment. The text was modified.

In line 137, does it mean 50mg of PRX 'dissolved' (not solved) in 9.95g of ethyl acetate? Please clarify/correct.

Thank you for your comment. The text was modified.

In line 164, was the pump speed set at 50ml/min? Please clarify.

Thank you for your comment. The pump speed has a dimensionless unit using NTA.

In line 305, Figure 5, upon closer observation, the red-colored PRX_POL_LIO (bottom), Piroxicam (Top), and POL (2nd), the last one seems not a physical mixture of PRX+POL; it appears to be a polymorph for me. My question that needs clarification here is: Did the solubility increase due to this new form, or due to nanosizing with Lyophilization? Check the literature for any existing PRX polymophs matching with the current new pattern in the figure to make sure it is not a new polymorph. If it is any existing polymorph then you have to explain that properly in the manuscript.

Thank you for your comment. The last one is not the physical mixture of PRX and POL. It is the freeze dried sample. The names of the samples have been clarified. The peaks, which can be seen in the case of PRX_POL_LIO, are belong to MAN. This was corresponding to the DSC results. The original anhydrate form of the PRX become amorphous during nanosizing and freeze drying, which is responsible for the solubility increase.

Reviewer 2 Report

Comments and Suggestions for Authors

This article designs an oral dispersible freeze-drying agent composed of nanoscale piroxicam (PRX), which improves drug absorption rate and oral bioavailability by reducing the particle size of PRX. This study characterized the morphology and structure of the formulation, and measured its in vitro disintegration time and drug release. The results indicate that this is an effective delivery method for non-steroidal anti-inflammatory drugs.

However, there are still some questions here.

(1)  In this paper, only the in vitro drug release of the preparation was measured, and the in vivo drug release was not measured. It is suggested that in vivo animal experiments should be performed to determine the blood concentration-time curve.

(2)  In structural characterization, it is recommended to supplement IR spectroscopic measurements.

(3)  Please proof-read the manuscript again as there are some typo errors and please check the reference format.

Comments on the Quality of English Language

Please proof-read the manuscript again as there are some typo errors and please check the reference format.

Author Response

Thank you very much for your opinion. Below are listed all of the modifications made in the paper, according to the suggestions, and you can find them with orange color in the text.

In this paper, only the in vitro drug release of the preparation was measured, and the in vivo drug release was not measured. It is suggested that in vivo animal experiments should be performed to determine the blood concentration-time curve.

Thank you for your valuable feedback. We appreciate your suggestion for in vivo animal experiments to determine the blood concentration-time curve. The in vivo investigations have been planned for the near future. However currently, the circumstances aren't ideal for carrying out the recommended in vivo tests. It requires an extensive amount of financial resources because we don't have the infrastructure for animal testing. Moreover, it's important to note that in vivo investigation falls outside of the main scope of this journal.

In structural characterization, it is recommended to supplement IR spectroscopic measurements.

Thank you for your comment. Fourier-transform infrared spectroscopy investigations was added to the manuscript.

2.7. Fourier-transform infrared spectroscopy (FT-IR)

The interactions between PRX and the excipients were investigated by the AVATAR 330 FT-IR spectrometer (Thermo Nicolet, Thermo Fisher Scientific Inc., Waltham, MA, USA). The raw materials were combined into physical mixtures (PM) having the same composition as OL. The samples were homogenized with 150 mg of KBr in an achate mortar, and the mixture was pressed to make pastilles using a Specac® hydraulic press (Specac, Inc., USA) by 10 ton pressing force. The infrared spectra were recorded between 4000 and 400 cm-1, at an optical resolution of 4 cm-1.

3.7. Fourier-transform infrared spectroscopy

FT-IR analysis was carried out to study the possible molecular interactions be-tween PRX and the excipients (POL, MAN, NaAlg). To identify the changes occurring during nanonization and freeze drying, the FT-IR spectra of the original PRX and MAN, the PM and the OL were compared (Figure 6). However, the ratios of the mate-rials led to that in the case of PM and OL, the PRX was not well observable due to the presence of MAN dominated the spectra. The shifts of the spectra of MAN after lyophilization could be the change of its structure[33], as reported by DSC and XRPD.

Figure 6. FTIR results of two initial materials, the PM and the OL.

FTIR spectra may indicates some interactions between the components, but it requires further investigations.

Please proof-read the manuscript again as there are some typo errors and please check the reference format.

Thank you for your comment. We reviewed the manuscript again to correct any typographical errors and ensure the reference format is accurate.

Round 2

Reviewer 2 Report

Comments and Suggestions for Authors

This revised version can be acceptable.